# *Eucalyptus* Short-Rotation Management Effects on Nutrient and Sediments in Subtropical Streams

**Carolina Bozetti Rodrigues** [1,*], **Ricardo Hideo Taniwaki** [2], **Patrick Lane** [3],
**Walter de Paula Lima** [4] **and Silvio Frosini de Barros Ferraz** [4,*]

1 Prática Socioambiental, Guararema 08900-000, Brazil
2 Centro de Engenharia, Modelagem e Ciências Sociais Aplicadas, Universidade Federal do ABC, Santo André 09210-580, Brazil; ricardo.t@ufabc.edu.br
3 School of Ecosystem and Forest Sciences, The University of Melbourne, Melbourne 3010, Australia; patrickl@unimelb.edu.au
4 Department of Forest Science, Luiz de Queiroz College of Agriculture (ESALQ), University of São Paulo (USP), Piracicaba 13418-900, Brazil; wplima@usp.br
* Correspondence: cabreuva@gmail.com (C.B.R.); silvio.ferraz@usp.br (S.F.d.B.F.); Tel.: +55-19-3447-6692 (S.F.d.B.F.)

**Abstract:** Forested catchments generally present conserved aquatic ecosystems without anthropogenic disturbances; however, forest management operations can degrade these environments, including their water quality. Despite the potential degradation, few studies have analyzed the effects of forest management in subtropical regions, especially in forest plantations with intensive management, such as *Eucalyptus* plantations in Brazil. The intensive management of those plantations is characterized by fast-growing, short rotation cycles, and high productivity. This study aimed to assess the effects of *Eucalyptus* plantations harvesting on the concentration and exportation of nutrients and suspended solids in subtropical streams. Results showed that clear-cut harvesting and subsequent forest management operations do not alter most of the concentration of nitrate, potassium, calcium, and magnesium. The concentration of suspended solids increased during the first year after timber harvesting in all studied catchments, however, the increases were statistically significant in only two catchments. In the first year after harvest, it was observed an increment of water yield/precipitation ratio at three catchments, which also increased export of nutrients and suspended solids. Our results showed that harvesting of fast-growing *Eucalyptus* forest plantations partially affected sediment exports and did not compromise water quality in the studied catchments. However, the catchment land-use design, especially related to road density and land-use composition, showed significant relationship with sediment exportation.

**Keywords:** timber harvesting; forest operations; nutrient concentrations; load; water quality

## 1. Introduction

Water quality of a stream is determined by a range of current and historical influences on catchment, from natural or anthropogenic origin, and is an important indicator of aquatic ecosystem health [1]. Streams draining forested landscapes usually have higher water quality than streams draining other land uses, such as agricultural fields [2–8]. The high quality of water provided by forested landscapes is partly attributed to a better soil infiltration and a variety of physical and biogeochemical ecosystem processes in the soil that filter particles and chemicals from the water [9].

From 1990 to 2015, native forests areas around the world were reduced from 4.28 to 3.99 billion of hectares [10]. On the other hand, forest plantation areas increased from 167.5 to 277.9 million

of hectares [10]. Forest plantations play an important role in providing roundwood for industrial and energy generation in several countries around the world [10,11]. In the last decade, forest plantations accounted for one-third of the world's industrial demand for roundwood and projections indicate that in 2040, half the world's demand for this type of raw material will be supplied by forest plantations [11,12].

In Brazil, 5.7 million hectares are occupied by *Eucalyptus* plantations [13]. Most of these forests are under intensive management, with short rotation cycles (from 6 to 8 years) and presents one of the highest productivity in the world (from 25 to 60 $m^3$ $ha^{-1}$ year $^{-1}$) [14]. As a consequence of intensive management, harvesting and other forest management operations can cause shifts in the concentrations and export rates of nutrients and solids, altering the water quality on streams [1–4,9,15–17].

Although extensive reviews have addressed the effects of forest management on water quality in different locations, most of these reviews have compiled studies from temperate regions, such as North America and other temperate regions of the world, or in specific countries such as New Zealand and Australia [1–4,7,9,15,17–22]. However, studies or reviews analyzing the effects of forest plantation management on water quality in tropical and subtropical ecosystems are still rare [2,23].

In tropical regions, the impact on water quality is expected to be higher due to intensive forest management and the use of large amounts of fertilizers [2,23]. In addition, tropical and subtropical regions have more intensive hydrological and biogeochemical cycles due to higher temperatures compared to temperate regions, resulting in larger variations in streamflow and nutrient exports [24]. Short rotation management in Brazil usually keeps soil exposed during a period after harvesting and subsequent operations (e.g., residue management, soil preparation, fertilization, liming etc.) [14] which increases the chances of nutrients and solids being transported to streams during rainfall events. In addition, soil infiltration capacity may be impaired by some machine operations involved in harvesting, resulting in compacted soil areas [25].

The main sources of suspended solids in forestry operations are roads and harvested areas. Intensive forestry operations depend on the construction and maintenance of roads which allow access to silvicultural areas. However, unpaved roads are the major source of sediment loads from harvested catchments due to erosion and runoff [20,22,26,27].

Forest cover removal by harvesting usually reduces evapotranspiration and precipitation interception, increasing the water reaching the soil at catchments [18,28–30]. Thus, even if there are no changes in nutrient and suspended solids concentrations, usually the exported values would increase due to the higher amount of water delivered to streams.

In this study we assessed effects of the intensive management of *Eucalyptus* plantations on the concentration and export of nitrate, potassium, calcium, magnesium, and suspended solids in subtropical catchments, aiming to understand effects of forest harvesting on water resources. Forest management characteristics are discussed in order to understand how they could attenuate or increase observed effects.

## 2. Materials and Methods

### 2.1. Study Areas

In this study, we select four catchments located in the state of São Paulo, southeastern Brazil (Figure 1). Catchments are located in private company areas, which were managed for commercial purposes, following usual short rotation forest operations, inherent to the pulpwood production process. The catchments were studied for two years, as follows: One year before forest harvesting (BH) and one year after harvesting (AH).

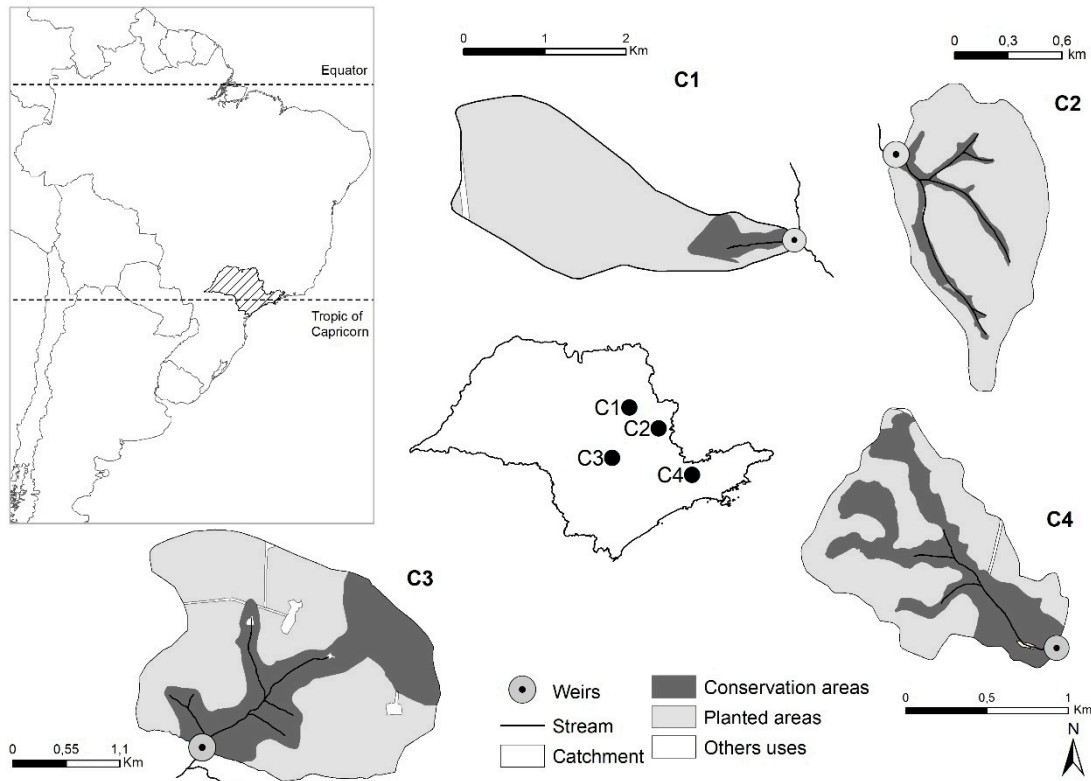

**Figure 1.** Location of the four catchments (C1, C2, C3 and C4) in the state of São Paulo, Southeast Brazil.

According to Köppen's climatic classification for Brazil [31], catchments C1 and C2 were classified as "Cwa" (humid subtropical with dry winter and hot summer), catchment C3 as "Cfa" (humid subtropical of oceanic climate, without dry season and with hot summer) and catchment C4 as "Cfa" and "Cfb" (humid subtropical of oceanic climate, without dry season and with temperate summer). The mean annual precipitation for the catchments was 1487 mm (C1), 1453 mm (C2), 1268 mm (C3), and 1425 mm (C4) [31].

Soil mapping of catchments was provided by forestry companies and classified according to United States Department of Agriculture (USDA) soil taxonomy as follows: Entisols Quartzipsamments, predominant in C1 (82%) and C3 (58%), and Inceptisols, predominant in C2 (64%) and C4 (59%) catchments (Table 1). Entisols Quartzipsamments are recently formed soils, freely drained, and they have more than 90% of resistant minerals; Inceptisols are young soils that commonly occur on landscapes where erosional processes are active, exposing unweathered materials [32]. Physical characteristics and land use proportions before and after harvesting of the catchments can be observed in Table 1. Immediately after harvesting, the land-use was modified in the catchments C1 and C2 by increasing conservation areas and reducing plantation areas. Forest harvesting was totally mechanized, and after that, a new forest was planted at the catchments C1, C2, and C3, while at C4 catchment, the growth of the shoots was carried out (coppice system).

### 2.2. Hydrological Data and Water Quality

A V-notch weir was built in the outlet of each catchment to collect discharge data. An automatic sensor (Table 2) was installed in each gauge station to measure and record the water level and precipitation (both at 15-minute intervals). The continuous water level records were used to calculate streamflow data by a specific equation developed for each weir. After that, annual water yield (Q) and annual precipitation (P) were calculated for BH and AH years.

**Table 1.** Main characteristics and distribution of land use on catchments C1, C2, C3 and C4.

| Characteristic | Catchments | | | | | |
|---|---|---|---|---|---|---|
| | **C1** | | **C2** | | **C3** | **C4** |
| Total area (ha) | 470.1 | | 86.6 | | 533.7 | 125.7 |
| Average slope (%) | 6.8 | | 14.3 | | 9.6 | 22.5 |
| Forest age (years) | 6 | | 7 | | 7 | 6 |
| Stream flow | perennial | | intermittent | | perennial | perennial |
| Main soil type (%) | Entisols[1] (82%) | | Inceptisols (64%) | | Entisols[1] (58%) | Inceptisols (59%) |
| Land use (%) | BH[2] | AH[3] | BH[2] | AH[3] | BH[2] | BH[2] |
| Forest plantation | 91.5 | 80.6 | 87.1 | 58.3 | 65.8 | 59.3 |
| Conservation areas | 7.6 | 18.4 | 12.7 | 41.5 | 32.5 | 39.9 |
| Others uses | 0.9 | 0.9 | 0.2 | 0.2 | 1.8 | 0.8 |
| Harvested area | 91.5 | - | 87.1 | - | 65.8 | 40.0 |
| Road density (m ha$^{-1}$) | 49.6 | 45.2 | 81.5 | 72.3 | 45.4 | 64.6 |

Note. [1] Entisols (Quartzipsamments); [2] BH = before harvesting; [3] AH = after harvesting.

**Table 2.** Harvesting dates, type of weirs, water level and rain gauge equipment, and number of water samples collected at catchments.

| Catchment | Harvesting Date | Type of Weir | Electronic Equipment | Water Samples | |
|---|---|---|---|---|---|
| | | | Water Level/Precipitation | BH[1] | AH[2] |
| C1 | 11/2009 | 90° | Campbell Scientific (models CS540 and CR510)/Texas Electronics (model TR525MR3) | 45 | 48 |
| C2 | 07/2008 | 50° | Campbell Scientific (models CS450 and CR500)/Hydrological Services | 15 | 30 |
| C3 | 10/2008 | 35° | Solinst (model 3001)/Solinst | 46 | 49 |
| C4 | 06/2009 | 20° | Campbell Scientific (models CS450 and CR510)/Hydrological Services | 51 | 52 |

Note. [1] BH = before harvesting; [2] AH = after harvesting

In addition, water samples were collected manually at a weekly frequency at the gauge station of each catchment during the previous year (BH) and the year after (AH) the forest harvest, covering all months of the year (except for months of intermittence of the catchment C2). The information about V-notch weir type, equipment, management information and the number of water samples collected each year are presented in Table 2.

Water samples were collected in 500 mL plastic bottles, after triple washing with water from the stream, and kept refrigerated until laboratory analysis. The concentrations of nitrate ($NO^{3-}$), potassium ($K^+$), calcium ($Ca^{2+}$), magnesium ($Mg^{2+}$), and total suspended solids (TSS) were determined. Nitrate concentrations were determined by the colorimetric method upon the reaction with brucine sulfate (APHA, 2005). Potassium concentrations were obtained by flame photometer (Flame Photometer Micronal B220, AJ Micronal, São Paulo, Brazil). Calcium and magnesium concentrations were determined by atomic absorption spectrophotometer (PerkinElmer Analyst 100, PerkinElmer, Waltham, MA, USA) and the concentrations of total suspended solids were obtained by differences of pre-weighted glass microfiber filters (Whatman GF/C, Merck KGaA, Darmstadt, Germany).

*2.3. Data Analysis*

Annual water availability of the catchments was assessed by the relation between annual water yield (Q) and annual precipitation (P) (Q:P). At an annual scale, the Q:P ratio is considered a key parameter to quantify the effects of land use changes on streamflow [8].

Catchments C2 and C4 presented nitrate concentrations below the detection limit (C2 one sample in AH; C4 seven samples in BH and five samples in AH). In these cases, half of the minimum detection limit corresponding to nitrate was considered [33]. The non-parametric test of Mann–Whitney was used to assess statistical differences in nutrient and total suspended solids concentrations between BH and AH years. The test was used after the verification of non-normal distribution and homoscedasticity

(Shapiro–Wilk and Levene tests, respectively) of the data. The differences were considered as significant when $p < 0.05$.

Nutrients and solids exported annually were calculated by the concentrations of nitrate, potassium, calcium, magnesium, and suspended solids and daily average discharge values. For this, nutrient and sediment concentrations of a given water sample were constant until the next water collected sample. The daily exports were integrated into the time of annual exports (kg year$^{-1}$) and divided by the corresponding area of each catchment (kg ha$^{-1}$ year$^{-1}$).

In order to assess the relationship between annual exports and characteristics of forest management, a correlation analysis was used to relate AH exports and, the density of roads (m ha$^{-1}$), harvested area (%) and native vegetation (%). The linear correlations were evaluated using the Ordinary Least Square (OLS) algorithm. Squared Pearson correlation coefficient ($r^2$) was calculated. All statistical analyses were performed in Past software (version 3.12, Øyvind Hammer, Natural History Museum, University of Oslo, Oslo, Norway).

## 3. Results

### 3.1. Catchment Water Availability

Catchments C1, C2, and C4 showed lower annual precipitation in the AH year compared with BH year (respectively, −33%, −17% and −9%). However, the Q:P ratio was higher in AH (respectively, 57%, 17%, and 72%) in relation to BH year, which means that a higher amount of precipitation was converted into streamflow in these catchments (Table 3). Catchment C3 showed higher precipitation at AH year compared to BH year (Table 3), however, both years presented precipitation below the regional climatic average of 1268 mm [31], and this fact may have contributed to the reduction of the Q:P ratio (−37%).

**Table 3.** Annual precipitation (P), water yield (Q) and Q:P ratio one year before (BH) and one year after (AH) forest harvesting at C1, C2, C3 and C4 catchments.

| Catchment | Precipitation (P) (mm) | | | Water Yield (Q) (mm) | | | Q:P | | |
|:---:|:---:|:---:|:---:|:---:|:---:|:---:|:---:|:---:|:---:|
| | **BH** | **AH** | **%**[1] | **BH** | **AH** | **%**[1] | **BH** | **AH** | **%**[1] |
| C1 | 1700.1 | 1138.6 | −33 | 130.7 | 137.0 | 5 | 0.08 | 0.12 | 57 |
| C2 | 1702.8 | 1414.2 | −17 | 253.1 | 244.9 | −3 | 0.15 | 0.17 | 17 |
| C3 | 983.5 | 1124.1 | 14 | 126.3 | 90.4 | −28 | 0.13 | 0.08 | −37 |
| C4 | 1576.5 | 1428.3 | −9 | 415.1 | 648.1 | 56 | 0.26 | 0.45 | 72 |

Note. [1] Differences, in percentage, between annual values obtained in BH and AH years.

### 3.2. Nutrient and Suspended Solids Concentrations

The lowest nitrate concentrations were recorded in C4 catchment and the highest concentrations in C1 catchment (Figure 2). C2 catchment showed the highest concentrations of potassium, calcium, magnesium, and total suspended solids (Figure 2), probably due to its intermittent flow characteristic. Lowest concentrations of potassium, calcium, and magnesium were recorded in C1 catchment and the lowest concentrations of total suspended solids in C3 catchment (Figure 2).

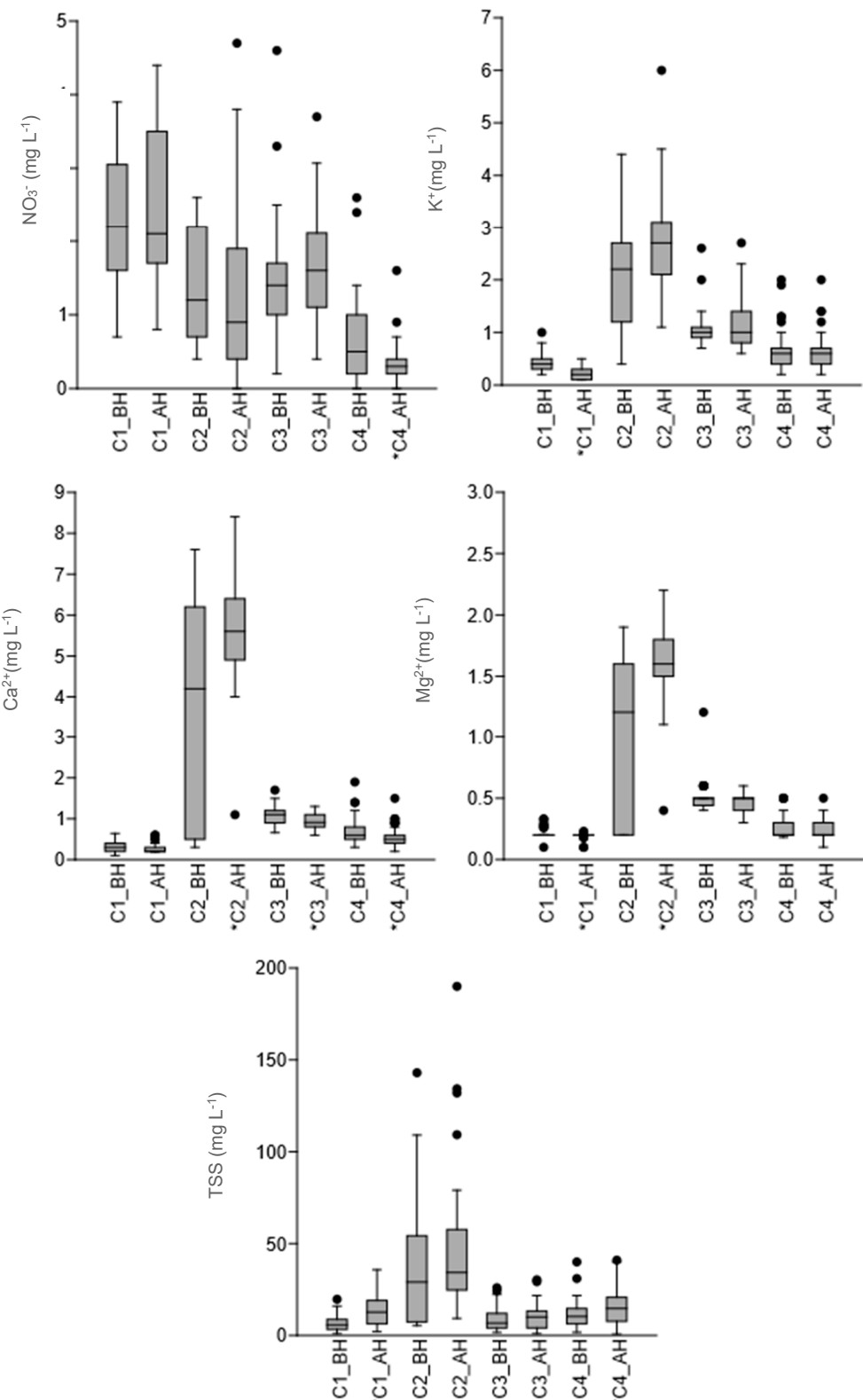

**Figure 2.** Concentration (mg L$^{-1}$) of nitrate (NO$_3^-$), potassium (K$^+$), calcium (Ca$^{2+}$), magnesium (Mg$^{2+}$) and total suspended solids (TSS) obtained one year before (BH) and one year after (AH) harvesting in catchments C1, C2, C3, and C4. * denote statistically significant differences between BH and AH years according to the Mann–Whitney test ($p < 0.05$). Outliers are represented by black dots.

The comparisons between nutrient and suspended solids concentrations in the year before (BH) and the year after harvesting (AH) showed significant differences ($p < 0.05$) in the studied catchments (Figure 2). Increment of magnesium (catchment C2), calcium (catchment C2), and suspended solids (catchment C1 and C4) were observed as effect of harvesting. Conversely, reduction after harvesting were observed for nitrate (catchment C4), potassium (catchment C1), calcium (catchment C3 and C4), and magnesium (catchment C1).

*3.3. Nutrient and Suspended Solids Exportations*

The catchment C2 presented increment of annual exports for all nutrients and suspended solids in AH year compared to BH year (Table 4). Although annual precipitation and water yield of the C2 catchment were lower (respectively, −7% and −3%) in AH year than in BH year, the Q:P ratio was 17% higher (Table 3). In contrast, the C3 catchment showed a reduction of exported nutrients and suspended solids in AH year compared to BH year (Table 4), following the trend of the annual values of water yield and Q:P ratio (Table 3).

**Table 4.** Exportation (kg ha$^{-1}$ year$^{-1}$) of nitrate (NO$^{3-}$), potassium (K$^{+}$), calcium (Ca$^{2+}$), magnesium (Mg$^{2+}$) and total suspended solids (TSS) one year before (BH) and one year after (AH) harvesting at catchments C1, C2, C3 and C4.

| Parameters | Exports by Catchment kg ha$^{-1}$ year$^{-1}$ (% of Change) | | | | | | | |
|---|---|---|---|---|---|---|---|---|
| | C1 | | C2 | | C3 | | C4 | |
| | BH[1] | AH[2] | BH[1] | AH[2] | BH[1] | AH[2] | BH[1] | AH[2] |
| Nitrate | 2.8 | 3.5 (22%) | 3.5 | 5.6 (59%) | 1.7 | 1.5 (−14%) | 2.9 | 2.5 (−14%) |
| Potassium | 0.5 | 0.3 (−44%) | 5.1 | 6.9 (36%) | 1.4 | 1.0 (−29%) | 2.7 | 3.8 (41%) |
| Calcium | 0.4 | 0.4 (0%) | 9.6 | 12.7 (33%) | 1.3 | 0.8 (−38%) | 2.5 | 3.3 (32%) |
| Magnesium | 0.3 | 0.3 (0%) | 2.6 | 3.8 (46%) | 0.6 | 0.4 (−33%) | 1.1 | 1.4 (32%) |
| Total suspended solids | 9.0 | 18.3 (104%) | 113.7 | 151.5 (33%) | 11.4 | 8.5 (−25%) | 49.8 | 106.3 (113%) |

Note. [1] BH = before harvesting; [2] AH = after harvesting

The C4 catchment showed higher exportation of most nutrients (except nitrate) and increment of suspended solids (Table 4). The C1 catchment showed annual exports increment only of nitrate (22%) and suspended solids (104%) (Table 4). C1 and C4 water yield and Q:P ratio were higher in AH year compared with BH year even considering the lower precipitation observed at after harvesting year (Table 3).

*3.4. Relationship between Forest Management and Exportation*

Significant positive relationships ($p < 0.05$) between road density and exports of potassium ($r^2 = 0.95$) and total suspended solids ($r^2 = 0.99$) were observed at the first year after forest harvesting (AH) (Figure 3). No relationship was found between nutrient exports and the percentage of areas of native vegetation or percentage of harvested area at studied catchments (Figure 3).

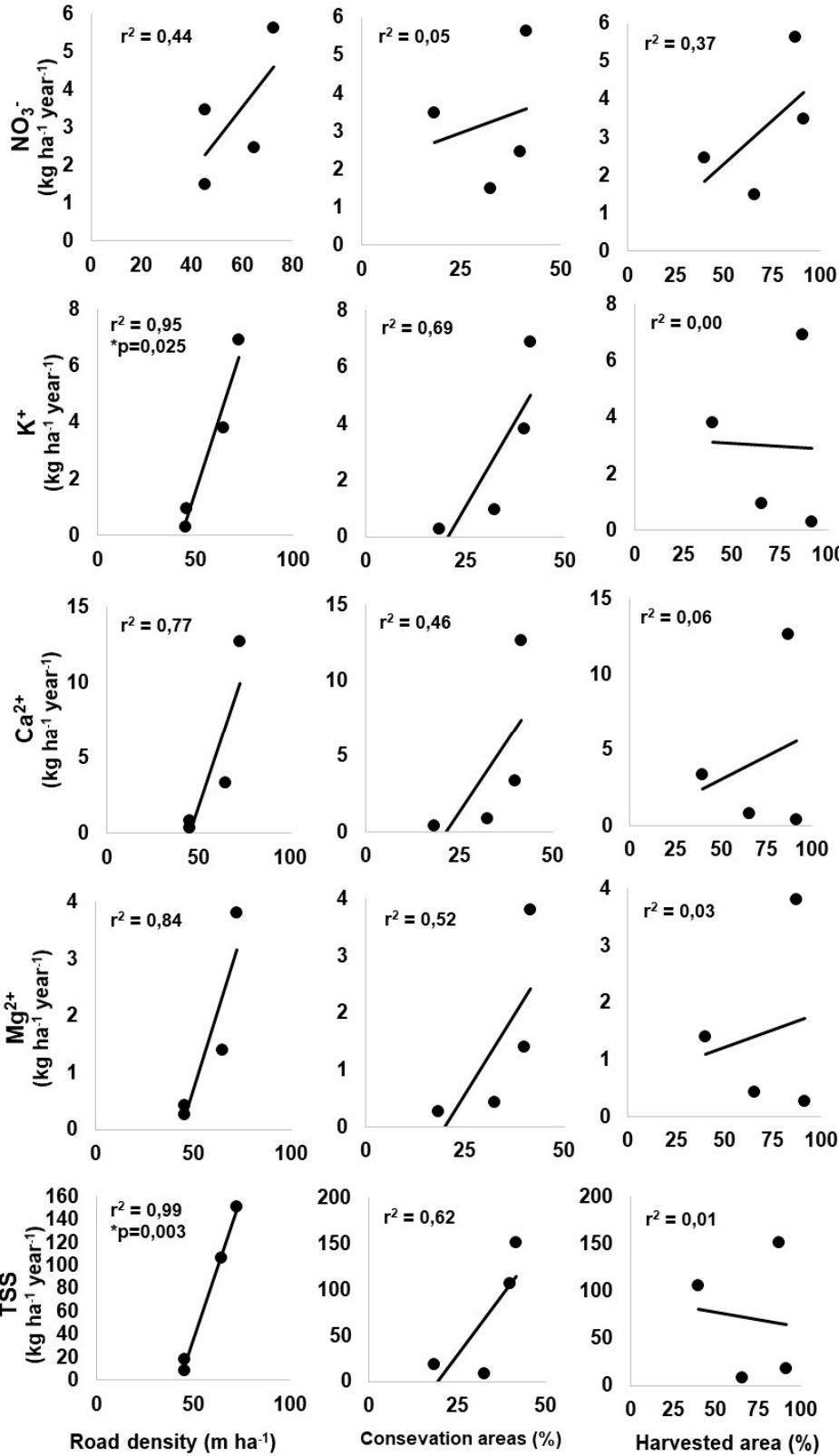

**Figure 3.** Exportation (kg ha$^{-1}$ year$^{-1}$) of nitrate (NO$^{3-}$), potassium (K$^{+}$), calcium (Ca$^{2+}$), magnesium (Mg$^{2+}$), and total suspended solids (TSS) obtained in the year after harvesting (AH) in relation to characteristics of forest management (road density, conservation areas and harvested area) of catchments C1, C2, C3, and C4. * denote relation statistically significant ($p < 0.05$).

## 4. Discussion

Nutrient and suspended solids concentrations observed in BH and AH years showed that the harvest and the subsequent forest operations did not alter most of the assessed parameters. These results are similar to those obtained in other studies showing that forest management operations in temperate regions partially change the water quality [2–4,16,17].

Results of this study also demonstrated that harvesting did not change nitrate concentrations in the catchments, in contrast to the results from temperate catchments [2,15,34]. The increment of nitrate concentrations after harvest can occur if the demand for nitrate by the vegetation is the dominant process that influences the presence of nitrate in stream water [17]. *Eucalyptus* plantations are characterized by a high demand for nutrients and high absorption capacity of soil nutrients [35–37], which could explain the stability of nitrate concentrations in the studied streams. In addition, it is estimated that in the first year *Eucalyptus* plantations demand 115 kg ha$^{-1}$ of nitrogen, 52 kg ha$^{-1}$ of potassium, 55 kg ha$^{-1}$ calcium, and 23 kg ha$^{-1}$ of magnesium [38], which would reduce the delivery of these nutrients to aquatic ecosystems.

Another major concern regarding the effects of forest operations on water quality is related to soil erosion and sedimentation [3,4,15]. All studied catchments showed an increment of suspended solids at AH year (statistically significant at C1 and C4 catchments), the same also observed by other studies [4,15,21,39]. According to [17], fast vegetation growth after harvesting is able to stabilize water quality parameters in a short time. Therefore, *Eucalyptus* plantations are extremely efficient, since the closure of tree canopies occurs between the first and second year of age, depending on the growth rate [14]. Considering that there are still no established ecological limits on acceptable changes in nutrient and solids concentrations due to silvicultural activities [3,40], in order to assess the effects on water quality it is recommended the stream monitoring before and after the activity (disturbance) in *Eucalyptus* plantations [17].

Nutrient exportation varied depending on the catchment characteristics but the increment of sediment exports was observed in all studied catchments. Increment of nutrient and suspended solids exports after harvesting have been described in several studies and they are usually attributed to streamflow increment in response interception and evapotranspiration reduction [3,4,18,41–43]. Soil compaction and less soil infiltration caused by mechanized equipment were also cited as responsible for streamflow increment [18,21,42].

Variations observed on the exportation of nutrients or suspended solids between catchments are expected to be different due characteristics of soils, rainfall, topography, and land use [4,7–9]. An example can be observed on C3 catchment, which showed opposite dynamics compared with other studies catchments: a reduction of exportation and water yield in BH year compared to AH year. In this case, a major factor is related to precipitation observed at C3 catchment, which was below the regional average.

Soil and relief could also explain variations observed on exportations of suspended solids results, which follow the order C2 > C4 > C1 > C3. Catchments C2 and C4 have a predominance of Inceptisols in their areas, which are characterized by shallow soils, whereas C1 and C3 catchments have a predominance of Entisols (Quartzipsamments), which are characterized by depth and well drained soils (sandy soils). Deep soils tend to better redistribute precipitation and reduce lateral flow generation in comparison to shallow soils [8]. In addition, C2 and C4 catchments present higher slope, and steeper terrains usually present higher rates of erosion and, consequently, sediment exports [9].

Comparing the amplitude of suspended solids exported by the four catchments in the studied period (values between 8.5 and 151.5 Kg ha$^{-1}$ year$^{-1}$), it is observed that these values are extremely low when compared to those previously found in the literature [18,21,44]. In São Paulo State it was observed annual exports of suspended solids before harvesting of 28.7 kg ha$^{-1}$ year$^{-1}$ and after harvesting, 60.6 kg ha$^{-1}$ year$^{-1}$, representing increment of 111% [39], similarly also observed by [45], where annual exports of suspended solids before harvesting was 19.8 kg ha$^{-1}$ year$^{-1}$ and after harvest, 41.5 kg ha$^{-1}$ year$^{-1}$ (increase of 110%). The same increment was observed at this study on catchments

C2 (104%) and C4 (113%). Exportation could be considered low when compared to studies conducted in the United States, were an increment of suspended solids ranging from 0% to more than 8000% were observed [7]. In Brazil, the maintenance of riparian vegetation around streams and springs is mandatory by law [46] and, for this reason, all studied catchments have conservation areas with native vegetation occupying from 7.6% to 39.9% of catchment. Several studies highlight the importance of riparian buffer strips along streams as a practice that can effectively mitigate the effects of forest operations on water quality [1,6,15–18,21,47,48]. This fact demonstrates the importance of intensifying studies on *Eucalyptus* plantations management in Brazil, in order to understand its effects on water quality.

Regarding the characteristics of forest management influencing nutrient exports, road density was related to potassium and suspended solids exports in AH year. Several studies have demonstrated roads effects on suspended solids at forest management areas, being considered as a permanent source of suspended solids for streams [4,9,15,21,26]. The descending order of suspended solids exports (C2 > C4 > C1 > C3) is coincident to the road density order observed of catchments. Therefore, road density reduction could contribute to the reduction of suspended solids delivered to the streams [26,49].

Changes in drainage patterns caused by roads reduce infiltration and, at steep areas combined with soil runoff, could increase the potential for detachment and transport of solids [4]. The infiltration capacity of roads is usually low and the lack of maintenance may result in higher runoff and sediment exports during precipitation events [26]. Solid particles can also transport nutrients adsorbed to them directly into the streams, increasing the amount of these nutrients exported [4], which justifies the positive relations between road density and exports of potassium, calcium, and magnesium. The absence of a linear relationship between nutrient exports and the percentage of harvested area has already been detected in some studies [34,50], being justified by the wide variety of factors and processes which control the dynamics of nutrients and suspended solids in streams.

Although not significant, the relationship between the proportion of native vegetation at catchment and exportation of nutrients and solids in streams was positive in the catchments studied. Besides this result seeming inconsistent, road density is related to the percentage of native vegetation, which means that the decreasing order of road density of catchments (C2 > C4 > C3 > C1) is the same for the percentage of areas for conservation. This fact is common in Brazil since the construction of roads around conservation areas is widely adopted for the prevention of forest fires. The prevention of forest fires is an indisputably important practice, but in some regions of the country this practice could be applied punctually and not in a generalized way, due to the low occurrence of fires. Although the areas set aside for conservation, mainly those located around the rivers and springs, have an important role in the protection of the water resources and the density of roads can counteract this function, diminishing its effectiveness.

## 5. Conclusions

This study demonstrated that Eucalyptus forest plantation harvesting shifts the concentrations and exports of nutrients and sediments in streams. The magnitude of these alterations can be aggravated or attenuated by natural characteristics of the catchments, such as type of soil and slope, and forest management choices, such as road density and land-use planning. Effects reduction depends on the adjustment of forest management to local physical characteristics.

**Author Contributions:** C.B.R. was responsible for data analysis and wrote the initial draft. R.H.T. helped to draft and edit the manuscript. S.F.d.B.F. and P.L. participated in the structuring of the manuscript and supervised the study. W.d.P.L. conceived the study. All authors contributed to the revision and edition of the manuscript.

**Funding:** This research was supported by "Fundação de Amparo à Pesquisa do Estado de São Paulo" (FAPESP), grants number: 2013/13243-9 and 2015/10502-9; and by "Coordenação de Aperfeiçoamento de Pessoal de Nível Superior" (CAPES).

**Acknowledgments:** The data set in this paper is part of a cooperative research program between Forestry Science and Research Institute (IPEF/PROMAB) and Brazilian forest companies. We thank all the employees of the participating companies for their valuable contribution. We are grateful for Lara Gabrielle Garcia who elaborated the map with the catchments.

**Conflicts of Interest:** The authors declare no conflict of interest.

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
