# Peer review of "Eucalyptus Short-Rotation Management Effects on Nutrient and Sediments in Subtropical Streams"

_forests, doi:10.3390/f10060519_

Round 1
Reviewer 1 Report
After
carefully checking the manuscript, I found the manuscript is generally
well organized, however, there some flaws throughout the manuscript. I
think methodology should be described in more detail and most figures
should be expressed more efficiently. Also, written English should be
double-checked by native speaker. I give my detailed comments as below:
Abstract is too long => first four sentences should be incorporated shortly
For introduction, road building effects on harvesting should be described in the intro.
IN 16; higher is not proper expression
IN 17: shift => degrade
IN 41: regions => areas
IN 84: inserted => included
IN 94 => 30 yr avg. annual precip. ?
In methodology, soil characteristic including soil texture should be described !
lN 106: water sampling => grab sample or automated sampling ?
IN 148: what is Past version ?
In table 3 => , => . this should be applied for all numbers
Figure 2=> caption should be expressed properly, ex) NO3-, K+
I think analysis related to figure 2 should be time series analysis rather than simple before/after comparison and the finding gained from the time series analysis can be described in discussion section.
What black dots mean in the figure 2 ?
Table 4: what does kg ha-1 year-1 / (% of change) mean ?
lN 196: subtitle is not proper. what about forest management activities instead of chrac. of forest management. How forest management affects the nutrients exports ?
Figure 2 & 3: each term(nutrient name) used in Y-axis should be expressed consistently.
In conclusion: you only give soil type not soil texture which is important factor associated with the exports.
Author Response
(a point-by-point response to the reviewer's comments was upload)

Reviewer 2 Report
The study is devoted to the actual problem of assessing changes in the concentration of nitrate, potassium, calcium, and magnesium in water bodies, as well as suspended solids due to deforestation of eucalyptus forests. The authors showed that the concentrations of chemical elements do not change, and the concentration of suspended substances increases in the first year after deforestation.
The authors' conclusion on the change in the amount of precipitation and water discharge of the watercourses before deforestation and after its deforestation is very interesting.
The following should be given as a comment:
1. Authors are advised to cover a longer period of observations (more than 2 years). This will allow you to determine how long the precipitation and water flow will return to the original figures that existed before deforestation.
2. It is necessary to describe in more detail the methods for measuring the amount of precipitation and water flow in the watersheds under study.
Author Response

(The authors gave the same response as above.)

Round 2
Reviewer 1 Report
After carefully checking the revised manuscript, the authors were successfully revised the manuscript as pointed out from the reviewer.
Now the manuscript looks much better in many aspects than it's initial impression.
However, there are still some places to be revised in written English and this should be double checked before publication.
Author Response
Dear Forests Editors and Reviewers,
We are very grateful for your reviews and comments on our manuscript “Eucalyptus short-rotation management effects on nutrient and sediments in subtropical streams”.
All observations and suggestions for improvements were duly incorporated into the manuscript. In addition, written English was carefully revised.
We hope that you could consider the manuscript reviewed and we are available for further information.
Please, feel free to contact us should you have any questions.
Sincerely yours,
Dra. Carolina Bozetti Rodrigues
Prof. Dr. Silvio Ferraz
